# Monitoring risk assessment on an acute psychiatric ward: Effects on aggression, seclusion and nurse behaviour

Esther J. R. Florisse[1]*, Philippe A. E. G. Delespaul[1,2]

**1** Mondriaan Mental Health Care, Heerlen, The Netherlands, **2** Department of Psychiatry and Psychology, School of Mental Health and Neuroscience, Maastricht University Medical Centre, Maastricht, The Netherlands

* e.florisse@mondriaan.eu

## Abstract

Evidence of risk assessment procedures is scarce and inconclusive. The aim of this study is to evaluate the effects of risk assessment on aggression and the use of coercive interventions in an acute psychiatric admission setting. In addition, we evaluated nurse behaviour before and after the use of risk assessment. To take the fluctuations with regard to aggression and coercive interventions into account, we allowed 26 weeks for baseline measurements, followed by a 26 weeks steady-state period after the implementation of the risk assessment instrument. Contrary to expectations, no positive effects of risk assessment were found on aggression or on coercive interventions. Time spent in seclusion increased significantly with more than 10 hours on average after implementation. Furthermore, there were only negative effects on nurse behaviour and experiences. Among other things, they felt more stressed, spent more time on administration tasks and spent less time with patients after the implementation. In conclusion, there is insufficient evidence to use structured short-term risk assessment to reduce aggression or coercive interventions.

## Introduction

Aggression and violence in psychiatric wards remain a major problem [1, 2]. The use of risk assessment tools in inpatient psychiatric settings may help to manage this problem [3, 4]. Risk assessment tools are developed to monitor the possibility of aggressive behaviour in individual patients and allow the staff to anticipate problems and make decisions to adjust management strategies and prevent further escalation.

Risk assessment is no panacea. One of the main criticisms is that it is useless when no reasonable intervention exists or is available to reduce future harm [5]. It is questionable if the staff will actually adjust their routines to prevent further escalations on basis of the risk assessment measurements. Also, there is little evidence for the effectiveness of risk assessment in reducing aggression. Accordingly, the ethical dilemma has been raised why to identify high-risk patients and treat them differently when risks are not actually prevented [5–9].

**Data Availability Statement:** All relevant data are within the manuscript and its Supporting Information files.

**Funding:** The author(s) received no specific funding for this work.

There are four published controlled studies that assess the use of risk assessment tools to reduce harm and aggressive incidents in different settings. Abderhalden et al. [10] and Van de Sande et al. [11] found a reduction in violence and aggression incidents when risks are monitored systematically on wards for acute inpatients. They also found a reduction in (the duration of) coercive interventions. In contrast, Kling et al. [12] found no beneficial effects in an acute hospital setting. Also, Troquette et al. [13] found no effects of risk assessment on violent or criminal behaviour in an out-patient forensic setting. None of these studies focussed on changes in staff behaviour towards patients. In short, the evidence for the use of risk assessment tools is inconclusive. Viljoen et al. [14] state in their systematic review that there is need for additional research.

This pilot study is an extension of the study by Van de Sande et al. [11]. It uses the Crisis Monitor [15] to assess violence risks as well as changes of staff behaviour and experiences on an acute psychiatric ward. In study 1 we assess the impact of the implementation of the Crisis Monitor on aggression and coercive interventions. We hypothesized that risk assessment reduces (the severity of) aggression. We also predicted a reduction of the number and duration of seclusions. Since involuntary medication is considered less intrusive compared to seclusions, we assumed that its use will not be affected. The frequency of applying a specific one-on-one de-escalation protocol, was expected to rise as an alternative for seclusion or other coercive interventions. In study 2 we assessed behaviour and experiences of staff. We hypothesized that the nurses will spend more time on the ward floor with patients, reduce avoidance behaviour and be more present when there is tension and threat. These changes in routine practice by nurses are known to have a positive effect on the therapeutic milieu and in keeping the unit safe [16–18].

## Study 1: Aggression and coercive interventions

### Methods

**Participants.** Mondriaan is an integrated mental health organisation in the south of the Netherlands. It offers both in- and outpatient care. The acute psychiatric admission ward of Mondriaan has 22 beds for a catchment area of 420000 inhabitants. The study took place on this specific inpatient ward. Administrative data of all procedures on this ward during baseline (October 2016 –April 2017) and during the steady state post-intervention period (July 2017—January 2018), were stripped from identification data and filed anonymously. Data include use of involuntary medication, number and duration of seclusions, number and duration of specific one-on-one treatment protocols.

**Design.** The study assesses the impact of the introduction of a Crisis Monitor on risk, actual aggression, containment interventions and staff behaviour. Because the number of seclusions and aggression incidents vary considerably over time, depending on the changing case mix of the admitted patients on the ward, the baseline period was 26 weeks. As such, we hoped to level out the fluctuations. After the baseline period, the team implemented the Crisis Monitor as a risk assessment tool. The nursing staff were trained to apply the Crisis Monitor as a routine and focus on de-escalating practices. Monitoring was introduced over a period of 11 weeks with instruction and practice. After that period, implementation was successful and a steady-state was reached. The follow-up period called the post-implementation steady-state period also had 26 weeks and assessed a fully implemented intervention (see Fig 1).

**Intervention.** The team of the admission ward implemented the Crisis Monitor within its routine practice. The psychiatric nurses and doctors were trained to rate the risk assessment tools (see measures below) at the end of each shift (excluding the night shift). The ratings of the instruments were discussed every morning during multidisciplinary meetings. If necessary,

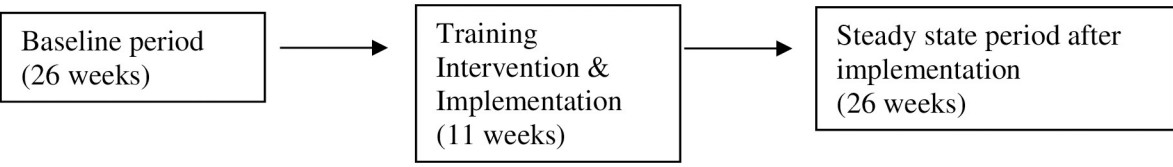

**Fig 1. Schematic timeframe of the study.**

decisions were made to manage risks of individual patients, by introducing de-escalating interventions. This was especially the case when the total score of the BVC was above cut-off (>2). Also, the team reviewed the daily and weekly ratings for each patient in more detail in the weekly care coordination meetings. This double strategy allows for short and long-term decision making. As in the study by Van de Sande and colleagues (2011), administration of the daily instruments of the Crisis Monitor took approximately 5 minutes, and the weekly version took about 15 minutes per patient (average investment with 20 patients: 1700 minutes/week). The team practised de-escalating strategies such as the use of the intensive care unit and offered one-on-one treatment.

**Measures.** The same instruments assessed the baseline and the steady state post-implementation period (see Table 1 for the assessment schedule).

*Crisis monitor.* The Crisis Monitor, developed by Van de Sande et al. [15, 19], is a risk assessment method which combines five standard observation scales (Kennedy short and long, BVC, BPRS, SDAS) to objectify the risk of violent behaviour in inpatient psychiatric settings.

*Kennedy-Axis V*: The Kennedy [20] has seven scales; i) psychological impairment, ii) social skills, iii) violence, iv) Activities of Daily Living (ADL)-Occupational skills, v) substance misuse, vi) medical impairment, vii) ancillary impairment. It generates a global functioning assessment and is an alternative to the DSM-IV-TR Axis 5 GAF Scale. In the Dutch version, an 8th scale ("motivation for treatment") is added [21]. The interrater reliability for the Dutch version assessed by nurses was 0.79 [22]. The short version of the Kennedy-Axis V consists of the first four daily-assessed domains. Rating of the extended version occurs weekly.

*BVC (Brøset Violence Checklist)*: The BVC [23] was also assessed twice per day (after each shift except the night shift) and aims to predict violence within the next 24-hour period. Each of the six items (i) confusion, ii) irritability, iii) boisterousness, iv) physical treat, v) verbal treat, vi) attack on objects) is scored as present or absent. It has a sensitivity of 63% and a specificity of 92% with the suggested cut-off score of 2. The Kappa value for the total BVC score was 0.44.

**Table 1. Measures during baseline and steady state period.**

| | Assessment moment | Baseline | Steady-state post-implementation |
|---|---|---|---|
| Crisismonitor | | | |
| • Kennedy-Axis V (short and long version) | day, week | | X |
| • BVC | day | | X |
| • BPRS | week | | X |
| • SDAS | week | X | X |
| Argus | | | |
| • Digital registration coercive interventions | event | X | X |
| • Digital registration one-on-one treatment | event | X | X |

Note. BVC = Brøset Violence Checklist, BPRS = Brief Psychiatric Rating Scale, SDAS = Social Dysfunction and Aggression Scale.

*BPRS (Brief Psychiatric Rating Scale)*: The BPRS, initially developed by Overall et al. [24, 25], consisted of 16 items. The scale focusses primarily on inpatient psychopathology and is particularly suitable for psychotic patients. The BPRS was modified and expanded. The Crisis Monitor uses the BPRS version of Bigelow et al. [26]. It is a 26 item scale rated from 1 to 7 and was measured on a weekly basis. Factor analysis yields 5 clusters; "positive symptoms/thought disorder", "depression", "negative symptoms/anergia", "mania", "disorientation" [27].

*SDAS (Social Dysfunction and Aggression Scale)*: The primary outcome measure is the type and severity of aggression, recorded with the SDAS [28]. This instrument was completed every week (or at the day of discharge) for each patient admitted longer than three days. The SDAS consists of 9 items covering outward aggression and two questions (9 and 11) that assess inward aggression: i) non-directed verbal aggression, ii) directed verbal aggression, iii) irritability, iv) negativism, v) dysphoric mood, vi) socially disturbing behaviour, vii) physical violence to personnel, viii) physical violence to others, ix) self-mutilation, x) physical violence to things, xi) suicidal thought and impulses). Assessors rated the SDAS on 5-point scales. Inter-observer reliability is 0.97. Cronbach's alpha coefficient of SDAS-9 (outward aggression) was 0.79, and the Loevinger coefficient of homogeneity was 0.40.

Summary scores for the SDAS are verbal aggression (item 1 and 2), physical aggression (scale 7, 8 and 10), auto-aggression (item 9 and 11) and psychological precursors related to aggressive behaviour (items 3, 4, 5 and 6). A total severity of aggression score sums all non-auto-aggression scales.

*Argus and systematic digital registration of coercive interventions.* Episodes of seclusion were recorded using the Argus scale which provides information about the incidence and duration [29, 30]. A seclusion incident is a sequence of periods of seclusion separated by no more than 24 hours. Besides this, the systematic digital registration of separation and seclusion on the psychiatric wards was used to evaluate the administration of involuntary medication and the number of specific one-to-one treatment protocols on the intensive care unit (ICU), in the proximity of the psychiatric ward.

**Statistical analysis.** Although Van de Sande et al. [11] did report data regarding the use of the Crisismonitor, a classic power analysis was not useful because the N is mainly influenced by unpredictable fluctuations of seclusions and aggression incidents over time. That is why we maximised the duration of the baseline and post-implementation steady state period to feasible limits (both 26 weeks). Differences in patient and nurse characteristics between the baseline and new steady state period were tested by chi-squared and *t*-tests. Because some subjects were admitted both in the pre- and post-phase, and therefore data were nested within subjects, multi-level regression analyses were used to assess the effect of the intervention on aggression and coercive interventions.

All analyses were performed with Stata (version 14; Stata corp., College Station, TX, USA) function xtreg (for ordinal) or xlogit (for nominal variables). The significance level for all statistical tests was set at $p < .05$, two-tailed.

**Study ethics.** This study was in accordance with the legislation and ethical standards on human experimentation in the Netherlands and in accordance with the Declaration of Helsinki (amended version 2013). The study was approved by the Commission for Scientific Research of the Mondriaan Mental health Trust in Heerlen/Maastricht, the Netherlands.

## Results

Inclusions amounted to 293 admissions (224 individual patients) during baseline and 252 admissions (208 individuals) during the post-implementation steady-state period. Thirty-five individual patients were admitted in both periods (see Table 2 for an overview of patient

**Table 2. Patient characteristics during baseline and post-implementation steady-state period.**

| | baseline | post implementation steady-state | statistical test | p |
|---|---|---|---|---|
| Number of admissions, $n$ | 293 | 252 | | |
| Number of individual patients, $n$ | 224 | 208 | | |
| Number of beds, $n$ | 22 | 22 | | |
| Duration patient admission, mean days (sd) | 16.9 (19.4) | 17.8 (17.2) | $\chi^2 (1) = 0.35$ | 0.552 |
| Patient characteristics | | | | |
| • age, years: mean (sd) | 39.2 (12.8) | 38.7 (12.7) | $\chi^2 (1) = 129.69$ | 0.000 |
| • gender, male: $n$ (%) | 160 (54.6) | 133 (52.8) | $\chi^2 (1) = 0.00$ | 0.944 |
| • ethnic minority, $n$ (%) | 31 (10.6) | 27 (10.7) | $\chi^2 (1) = 0.03$ | 0.860 |
| • involuntary admitted, $n$ (%) | 214 (73.0) | 166 (65.9) | $\chi^2 (1) = 2.82$ | 0.093 |
| • diagnosis | | | $\chi^2 (3) = 0.81$ | 0.847 |
| psychotic disorder | 190 (64.9) | 162 (64.3) | $\beta = -0.15$ (0.15) | 0.333 |
| personality disorder | 47 (16.0) | 39 (15.5) | $\beta = -0.25$ (0.39) | 0.527 |
| drug misuse | 30 (10.2) | 30 (11.9) | $\beta = 0.21$ (0.38) | 0.581 |
| other primary diagnosis | 26 (8.9) | 21 (8.3) | $\beta = -0.03$ (0.44) | 0.942 |

characteristics). We found no significant differences for gender, ethnic minority, involuntary admissions or diagnosis between patients in both periods. Patients during the post-implementation steady-state period were younger than the patients admitted at baseline (z = -11.39, p < .01).

**Aggression.** With regard to aggression, our primary outcome measures, there were no differences between baseline and post-implementation steady-state period on any of the sub-scales of the SDAS (verbal aggression, $\beta = .03$, p = .72; physical aggression, $\beta = .08$, p = .07; auto-aggression, $\beta = .08$, p = .06; psychological precursors, $\beta = .08$, p = .39) nor on the total severity score ($\beta = .07$, p = .12). See Table 3 for an overview of the primary outcome measures.

**Coercive interventions.** The mean hours spent in seclusion increased significantly with more than 10 hours per episode after the introduction of the Crisis Monitor ($\beta = 27.05$, p = .03). The number of seclusion incidents did not differ between the two periods ($\beta = -.05$,

**Table 3. Outcome measures during baseline and post-implementation steady-state period, multilevel regression estimates.**

| | baseline | post implementation steady-state | $\beta$ (sd) | p |
|---|---|---|---|---|
| SDAS | | | | |
| number of aggression incidents, $n$ subscales | 558 | 577 | | |
| • verbal aggression, $M$ (sd) | 0.85 (1.41) | 0.90 (1.14) | 0.03 (0.09) | 0.722 |
| • physical aggression, $M$ (sd) | 0.22 (0.58) | 0.28 (0.58) | 0.08 (0.04) | 0.069 |
| • auto-aggression, $M$ (sd) | 0.17 (0.60) | 0.29 (0.71) | 0.08 (0.04) | 0.056 |
| • psychological precursors, $M$ (sd) | 1.12 (1.42) | 1.20 (1.14) | 0.08 (0.09) | 0.390 |
| total severity score, $M$ (sd) | 0.65 (0.85) | 0.73 (0.73) | 0.07 (0.06) | 0.199 |
| Seclusion | | | | |
| number of aggression incidents, $n$ | 73 | 75 | -0.01 (0.05) | 0.789 |
| average duration in hours, $M$ (sd) | 20.5 (48.5) | 30.9 (63.6) | 27.05 (12.14) | 0.026 |
| Involuntary medication, $n$ | 75 | 86 | 0.57 (0.05) | <0.001 |
| Specific one-to-one treatment | | | | |
| number of interventions, $n$ | 26 | 25 | -0.05 (0.04) | 0.226 |
| average duration in hours, $M$ (sd) | 38.2 (63.2) | 42.5 (42.5) | 4.29 (15.15) | 0.777 |

Note. SDAS = Social Dysfunction and Aggression Scale.

$p$ = .83). The use of involuntary medication significantly increased from 75 to 86 incidents ($\beta$ = .57, $p$ < .01).

There was no difference in the number ($\beta$ = -.05, $p$ = .27) nor in the duration ($\beta$ = 4.29, $p$ = .78) of de-escalating one-to-one treatments.

## Study 2: Staff behaviour and experiences

### Methods

**Participants.**   The second study assessed the experiences and behaviour of the nursing staff on the acute psychiatric wards of Mondriaan before and after the introduction of the Crisis Monitor. During baseline there were 38 nurses working on the ward, 19 men and 19 women, during post-implementation steady state there were 34 nurses working on the ward, 14 men and 10 women. All nurses on the acute psychiatric ward were asked whether they were willing to participate in the study. They all agreed to participate. Because only 2 nurses per shift were needed in the specific weeks of measurement, not everyone was included. A total number of 19 nurses (5 men, 14 women) were included during baseline and 22 nurses (5 men, 17 women) during post-implementation steady state period. All participants were fully informed and consent was collected.

**Design.**   (see study 1).

**Measures.**   *PsyMate*™. The Experience Sampling Method (ESM; [31]) was adapted to sample the staff behaviour and experiences in different situations. This information was registered by nurses using the PsyMate™ App. The PsyMate™ App is designed to monitor experiences and behaviour in daily life. It runs on smartphones or iPods, has a user-friendly interface and allows easy and flexible programming. At random moments during the day, nurses receive a signal on their device and are asked to respond to several questions in reference to the moment directly before the beep. Questions include: What are you doing (i.e. administration, in contact with patient, having a meeting), where are you (i.e. outside the ward, in room of patient, in shared communal areas) and with whom are you (i.e. alone, patient, colleagues). There were also questions about the experiences and feelings at that moment, scored on a 7-point Likert scale: "I feel relaxed", "I feel safe", "I feel stressed", "I feel in control", "I feel anxious", "I feel tired", "I feel cheerful", "I feel lonely", "I feel work pressure", "this beep disturbed me". During every 8-hour shift (3/day), 2 nurses registered information with the Psymate™. Each nurse received 6 random assessment moments per shift, leading to a maximum of 36 registrations each day. Each registration took less than 90 seconds. Registration days were clustered in blocks of 1 week (7days), once every 2 months, resulting in 21 assessment days during baseline and 21 in the post intervention steady state period (average investment: 48 minutes/week).

**Statistical analysis.**   The data are clustered within individuals and therefore not independent. Consequently, multilevel linear regression techniques were used to analyse the ESM data using the xtreg or xtlogit packages from Stata (version 14; Stata corp., College Station, TX, USA) with subject id as random intercept factor. The significance level for all statistical tests was set at $p$ < .05, two-tailed.

### Results

A total number of 19 nurses were included during baseline and 22 nurses during post-implementation steady state period. 15 of these nurses were included in both periods. In both groups 5 male nurses participated in the study. Mean age of the nurses was 35 years at baseline and 32.6 years during the post-implementation steady-state period. Working experience as nurse was 9.9 years in baseline and 6.9 years in post-implementation period. There were no significant differences in age ($\chi^2$ (15) = 5.21, $p$ = .99), gender ($\chi^2$ (1) = .07, $p$ = 0.79), school

**Table 4. PsyMate outcome during baseline and post-implementation steady-state period, multilevel regression estimates.**

| | baseline | post implementation steady-state | *β* (sd) | p |
|---|---|---|---|---|
| Number of nurses, n | 19 | 22 | | |
| Number of responses, n | 519 | 471 | | |
| PsyMate | | | | |
| What are you doing? *n* (%) | | | | |
| Administration | 135 (26.0) | 155 (32.9) | 0.540 (0.07) | <0.01 |
| In contact with patient | 156 (30.1) | 112 (23.8) | -0.03 (0.04) | 0.358 |
| Having a meeting | 64 (12.3) | 52 (11.0) | -0.06 (0.05) | 0.243 |
| Communication | 22 (4.2) | 23 (4.9) | -0.01 (0.07) | 0.859 |
| Other work related | 79 (15.2) | 74 (15.7) | -0.02 (0.04) | 0.622 |
| Other private time | 63 (12.1) | 55 (11.7) | -0.05 (0.05) | 0.304 |
| *Overall model* | | | $\chi^2 (5) = 2.06$ | 0.840 |
| With whom? *n* (%) | | | | |
| Alone | 74 (14.3) | 61 (13.0) | 0.47 (0.07) | <0.01 |
| Patient (alone) | 154 (29.8) | 113 (24.1) | 0.03 (0.05) | 0.451 |
| Patient (group) | 32 (6.2) | 39 (8.3) | 0.07 (0.06) | 0.254 |
| Close relatives | 1 (0.2) | 2 (0.4) | 0.38 (0.25) | 0.133 |
| Colleagues team | 244 (47.2) | 237 (50.5) | 0.06 (0.04) | 0.176 |
| Colleagues Mondriaan | 10 (1.9) | 17 (3.6) | 0.04 (0.09) | 0.689 |
| Colleagues Extern | 2 (0.4) | 0 (0) | -0.20 (0.31) | 0.514 |
| *Overall model* | | | $\chi^2 (6) = 4.68$ | 0.585 |
| Where are you? *n* (%) | | | | |
| Nursing post (counter) | 228 (44.1) | 247 (52.7) | 0.54 (0.07) | <0.01 |
| Consulting room | 105 (20.3) | 81 (17.3) | -0.07 (0.04) | 0.086 |
| Communal spaces ward | 59 (11.4) | 39 (8.3) | -0.07 (0.05) | 0.145 |
| Other rooms at ward | 19 (3.7) | 10 (2.1) | -0.11 (0.08) | 0.190 |
| Room patient | 23 (4.5) | 11 (2.4) | -0.11 (0.08) | 0.146 |
| ICU / Sep block | 34 (6.6) | 28 (6.0) | 0.07 (0.05) | 0.154 |
| Other | 34 (6.6) | 28 (6.0) | -0.05 (0.06) | 0.392 |
| Outside the ward | 15 (2.9) | 10 (2.1) | -0.07 (0.09) | 0.447 |
| *Overall model* | | | $\chi^2 (7) = 10.97$ | 0.140 |
| Feeling relaxed (L1-L7), mean (sd) | 5.92 (1.33) | 5.43 (1.26) | -0.56 (0.08) | <0.01 |
| Feeling safe (L1-L7) | 6.31 (1.08) | 6.05 (1.13) | -0.27 (0.07) | <0.01 |
| Feeling stressed (L1-L7) | 1.66 (1.14) | 2.01 (1.23) | 0.41 (0.07) | <0.01 |
| Feeling in control (L1-L7) | 6.21 (1.19) | 5.85 (1.16) | -0.36 (0.08) | <0.01 |
| Feeling anxious (L1-L7) | 1.30 (0.74) | 1.49 (0.77) | 0.22 (0.05) | <0.01 |
| Feeling tired (L1-L7) | 3.26 (1.90) | 3.78 (1.58) | 0.63 (0.11) | <0.01 |
| Feeling cheerful (L1-L7) | 4.99 (1.39) | 4.42 (1.44) | -0.73 (0.09) | <0.01 |
| Feeling lonely (L1-L7) | 1.37 (0.91) | 1.58 (0.97) | 0.25 (0.06) | <0.01 |
| Feeling work pressure (L1-L7) | 1.76 (1.23) | 2.35 (1.43) | 0.40 (0.08) | <0.01 |
| Feeling disturbed by beep (L1-L7) | 1.80 (1.72) | 3.05 (2.22) | 1.19 (0.13) | <0.01 |

level ($\chi^2 (1) = .03$, $p = .87$) or working experience ($\chi^2 (16) = 6.43$, $p = .98$) for both periods. 519 moments were documented with ESM during baseline period and 471 during post-implementation steady state period.

At random moments within the day nurses were asked what they did, where they were and with whom, and how they felt (see PsyMate™ summary data in Table 4). Overall, time spent on administration tasks increased with 26.5% (32.9/26), while time spent in contact with patients

alone decreased more than 19% after the implementation of the Crisis Monitor. Also, nurses were 19% more often in the nursing post, an open counter on the ward and 27% less present at the communal spaces at the ward, such as the living room, kitchen, etcetera. In regression analyses, the overall tests of time budgets (what the nurses were doing at the moment, with whom they were and where they were) were not significant ($\chi^2$ (5) = 2.06, $p$ = .84; $\chi^2$ (6) = 4.68, $p$ = .59; $\chi^2$ (7) = 10.97, $p$ = .14 respectively).

The PsyMate™ also assessed how nurses felt while at work. During the post-implementation steady-state period they were less relaxed and cheerful ($\beta$ = -.56, $p < .01$; $\beta$ = -.73, $p < .01$), they felt less safe and in control ($\beta$ = -.27, $p < .01$; $\beta$ = -.36, $p < .01$) and they felt more stressed, anxious and tired ($\beta$ = .41, $p < .01$; $\beta$ = .22, $p < .01$; $\beta$ = .63, $p < .01$). The experience of work pressure increased ($\beta$ = .40, $p < .01$) and they were more disturbed by the beeps of the PsyMate™ in new steady state period ($\beta$ = 1.19, $p < .01$).

To evaluate the objective level of stress during the two periods of the study a post-hoc regression was analysed in which for every day the objective stress level is calculated using minutes spent in seclusion, number of administered involuntary medication and use of coercive interventions outside office hours. The overall objective stress level was higher in the post-implementation steady-state period compared to baseline ($\beta$ = .43, $p < .01$). None of the subjective feelings measured with the PsyMate™ were related to the objective level of stress of that day (relaxed $\beta$ = -.01, $p$ = .81; safe $\beta$ = -.01, $p$ = .88; stressed $\beta$ = .01, $p$ = .78; in control $\beta$ = .03, $p$ = .44; anxious $\beta$ = .01, $p$ = .57; tired $\beta$ = -.003, $p$ = .95; cheerful $\beta$ = -.02, $p$ = .73; lonely $\beta$ = -.01, $p$ = .74).

## Discussion

Our study did not confirm the anticipated beneficial effect of the use of a structured short-term risk assessment tool in an acute psychiatric admission ward. Results showed no reduction of aggression nor in the use of coercive interventions. Time spent in seclusion increased significantly with more than 10 hours on average after the introduction of the Crisis Monitor in post-implementation steady state period when procedures were routine. These results contrast with the study of Van de Sande et al. [11]. They found beneficial effects on the reduction of aggression and time spent in seclusion using the same risk assessment tools as in the present study.

There are several possible explanations for these unexpected results. The patients differed between the control and experimental conditions. We tried to control for case-mix variation in admitted patients by using a 26-week window in the baseline and post-intervention steady-state period. We hoped this would have reduced potential differences between the groups. But it remains still possible that sampling fluctuations interfered with the results. It could be that during the post-implementation steady-state period a more difficult patient group was admitted than during baseline. This was the reason for a prolonged assessment period of 26 weeks. The strategy seems to have worked since there were no significant differences in patient characteristics in both periods, except for age. Moreover, the proportion of involuntarily admitted patients was lower in the post-implementation steady-state period. It is, therefore, improbable that the absence of positive effects is due to a more difficult patient group.

The teams in the studies by Van de Sande et al. [11] and Abderhalden et al. [10] were enrolled in a program to reduce seclusion and restraint and not blind for the condition. Expectancy and enthusiasm for non-coercive practices may have created non-intervention related differences between the wards. The staff of the present study was part of a nation-wide program to reduce coercive methods for some years and had already reduced these practices by 50% at baseline. The same team was involved in the pre and post phases, and changes were

related to historical factors as well as a higher threshold for seclusion. With an already low baseline, a reduction is more challenging to reach. Staff in the present study were used to focus on the precursors of aggression in their routine clinical observations and respond accordingly without the help of a structured risk assessment procedure. Differences in staff or ward culture, unrelated to the introduction of the Crisis Monitor, were more pronounced in the control conditions in the Van de Sande et al. study. Therefore, the present study potentially could not demonstrate a positive effect of the Crisis Monitor.

The same team was in charge at baseline and follow-up. After the implementation of the Crisis Monitor nurses spent more time on administration tasks, spent less time with patients and felt more stressed at follow-up. Filling in a risk assessment tool is an administrative task, so it is not surprising that the nurses spent more time on this after implementation. However, an increase of more than 25% is unexpected, and this raises the question whether it is acceptable that nurses in an acute psychiatric ward spent more time on administrative tasks while they should be in contact with patients. In fact, in the post-implementation steady-state period the nurses spent 19% less time with patients. The administrative burden due to study-related additional data collection (questionnaires and PsyMate™) was equal in the pre- and post-periods and could not account for the difference. Also, PsyMate™ ratings refer to the moment before the beep and is not part of the administration time. On itself the administrative burden of the risk assessment (5 minutes by patient for 2 shifts each day and 15 minutes for each patient per week) results in an extra administrative burden of 25 hours each week on a ward with 22 patients and an average occupancy rate of 80%.

To further explore the reasons for the present results, we interviewed the nurse practitioner and the head nurse. Both were intensively involved in the process of implementing the Crisis Monitor. They reported that feelings of anxiety, stress and work pressure often are related to team composition. The presence of inexperienced nurses or non-permanent team members with a flexible contract is considered a stressor with potentially detrimental effects on the mood, feelings and behaviour of the other nurses. The head of nurses stated that there were 9 nurses in the baseline period with less than one-year experience at the specific ward compared to 6 nurses in post-implementation steady state period. Unfortunately, there are no specific records of the exact team composition (i.a. substitution forces) at the time the study took place, so there was no opportunity to examine the possible differences in team composition between baseline and post-implementation steady-state period.

According to the nurse practitioner and head nurse, another stressor is the anticipation that coercive interventions outside of office hours, with lower staffing, would be needed. The administrative data contained records of the specific time of the use of coercive interventions. A posthoc analysis was performed to examine whether nurses had to use more coercive interventions outside office hours during post-intervention steady-state period. It appeared not to be the case for both seclusion and the use of involuntary medication ($\beta$ = -.12, p = .33; $\beta$ = .08, p = .65 respectively). Overall, on weekend days in which generally fewer nurses are present at the ward no more coercive interventions occurred than on other days (seclusion $\beta$ = 143.67, p = .38; involuntary medication $\beta$ = -.08, p = .75).

A final factor reported by the nurse practitioner and the head nurse was the presence of a specifically challenging patient during four months of the post-implementation steady-state period. This person behaved particularly violent, and on many occasions, police had to be called in to make sure the nurses and doctors were safe when face to face contact took place or during coercive interventions. The multi-level regression analysis accounts for high-frequency subjects and posthoc analyses with outliers removed, ruled out that this particular patient caused the unexpected finding. But it is possible that his behaviour caused a more stressful ward climate.

Kling et al. [12] and Troquette et al. [13] showed no beneficial effects of the use of structured risk assessment. They studied different settings (acute hospital setting and out-patient forensic setting respectively) and found the same results as the present study. Van de Sande et al. [11] and Abderhalden et al. [10] found a positive effect of risk assessment, but their studies have limitations. In conclusion, there is insufficient evidence to use structured short-term risk assessment to reduce aggression or coercive interventions. Potentially, the use of such a tool can negatively influence the experiences and behaviour of the nurse staff.

## Supporting information

**S1 File.**
(DTA)

**S2 File.**
(DTA)

**S3 File.**
(XLSX)

**S4 File.**
(XLSX)

**S5 File.**
(XLSX)

**S6 File.**
(XLSX)

**S7 File.**
(XLSX)

**S8 File.**
(DOCX)

## Acknowledgments

Our gratitude goes to all who helped to conduct this research: first of all, the nurses and clinical staff who cooperate so well and helped us to collect the data, with special thanks to Nina van Vlodrop, Paola Geijselaers, Romy van Es, Simone Bos, Nathalie Jonkers—Thijssen and Ilse van Goor. Karel Borkelmans for his support with regard to collecting the ESM data and Justine Lamée for taking over duties during leave of absence.

## Author Contributions

**Conceptualization:** Esther J. R. Florisse, Philippe A. E. G. Delespaul.

**Data curation:** Esther J. R. Florisse, Philippe A. E. G. Delespaul.

**Formal analysis:** Esther J. R. Florisse, Philippe A. E. G. Delespaul.

**Investigation:** Esther J. R. Florisse.

**Methodology:** Esther J. R. Florisse, Philippe A. E. G. Delespaul.

**Project administration:** Esther J. R. Florisse, Philippe A. E. G. Delespaul.

**Resources:** Esther J. R. Florisse, Philippe A. E. G. Delespaul.

**Supervision:** Esther J. R. Florisse, Philippe A. E. G. Delespaul.

**Validation:** Esther J. R. Florisse, Philippe A. E. G. Delespaul.

**Visualization:** Esther J. R. Florisse.

**Writing – original draft:** Esther J. R. Florisse, Philippe A. E. G. Delespaul.

**Writing – review & editing:** Esther J. R. Florisse, Philippe A. E. G. Delespaul.

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
