## [Decision Letter · Decision Letter 0]

3 Mar 2020

PONE-D-19-34513

Monitoring risk assessment on an acute psychiatric ward: effects on aggression, seclusion and nurse behaviour

PLOS ONE

Dear Drs. Florisse,

Thank you for submitting your manuscript to PLOS ONE. After careful consideration, we feel that it has merit but does not fully meet PLOS ONE’s publication criteria as it currently stands. Therefore, we invite you to submit a revised version of the manuscript that addresses the points raised during the review process.

Overall, the paper addresses an interesting and much debated topic. I have received two reviews from experts in your area of research. As you can see below, the comments are straightforward. Reviewer #1 has only minor comments. Also the comments from the second reviewer are relatively minor. S/he invites you to elaborate a bit more on the way staff was instructed etc. You should have no difficulties in addressing the comments from both reviewers.

We would appreciate receiving your revised manuscript by Apr 17 2020 11:59PM. To enhance the reproducibility of your results, we recommend that if applicable you deposit your laboratory protocols in protocols.io, where a protocol can be assigned its own identifier (DOI) such that it can be cited independently in the future. For instructions see: http://journals.plos.org/plosone/s/submission-guidelines#loc-laboratory-protocols

We look forward to receiving your revised manuscript.

Kind regards,

Robert Didden

Academic Editor

PLOS ONE

Journal Requirements:

3. Your ethics statement must appear in the Methods section of your manuscript. If your ethics statement is written in any section besides the Methods, please move it to the Methods section and delete it from any other section. Please also ensure that your ethics statement is included in your manuscript, as the ethics section of your online submission will not be published alongside your manuscript.

Reviewers' comments:

Reviewer's Responses to Questions

**Comments to the Author**

1. Is the manuscript technically sound, and do the data support the conclusions?

Reviewer #1: Yes

Reviewer #2: Partly

2. Has the statistical analysis been performed appropriately and rigorously? 

Reviewer #1: Yes

Reviewer #2: Yes

3. Have the authors made all data underlying the findings in their manuscript fully available?

Reviewer #1: Yes

Reviewer #2: Yes

4. Is the manuscript presented in an intelligible fashion and written in standard English?

Reviewer #1: Yes

Reviewer #2: Yes

5. Review Comments to the Author

Reviewer #1: This is an interesting study on a topic that is the subject of great uncertainty and debate. It not only demonstrates that quantifying the effectiveness of risk assessment in reducing risk remains contentious, challenging and questionable, it also illustrates the potentially detrimental impact that such practices have on clinicians and engagement with patients.

The design of the study and analysis of results appears sound, the discussion raises pertinent issues (especially in comparison with previous studies) and the associated limitations acknowledged.

If I could make some minor suggestions on grammar. Some of the tense is incorrect throughout the paper. See below.

Remove 'behavioural' after changes on line 62.

Also remove 'the' before patients (line 62) and before 'staff' (line 66). Same for line 74 (x2)

Under 'Design'

'This study assessed' not 'assesses'.

Remove 'long' after 26 weeks (line 97)

Line 98 "Nursing staff were trained"

Line 99, remove 'the' and start with "Monitoring....

Under intervention, I was confused by the word 'taxation' should this be 'assessment'?

Line 131, Suggest removing 'abuse' and replace with 'misuse'.

Line 137 "The BVC was" not 'is'

Line 145, suggest removing 'several times'.

Line 151, suggest changing 'filled in' with 'completed'.

Reviewer #2: Dear author

Thanks for a well written and relevant article. I have a few comments for you:

I need for you to elaborate a bit more on who you have instructed the staff to use the crisis monitor. In line 137 you write about BVC. The way you describe is ok, but I think there is a lack in your description of who to use the BVC. As I understand from your description the BVC is only scored once a day. The official manual formulates that the BVC is scored once each shift (that would in your case be three times a day). You are missing a reference to the results of BVCs capabilities.

I also need for you to describe how you have instructed the staff to act after a score above the cut-off score on the BVC. If you use the BVC a standalone Risk assessment tool there is an agreement that a score on 3 or above triggers an intervention – I can not read from your paper how you have instructed the staff about such thing. Because of these shortcomings I have some reservation about your conclusion. In risk-assessment research I think there is an agreement that the risk-assessment tool in it self is not enough it is the intervention therefore before I In your paper can see a description of how the staff is instructed to use there results og the score on the Crisis monitor I cannot accept the paper.

In your results from study 2 you have some significant results from baseline to post implementation “What are you doing” administration, “with whom” alone and “Where are you” Nursing post. I’m missing that you take into account that you ask the staff to both use a crisis monitor that increases the administration load and the you also disturb the staff with replying to the PsyMate – all in all that in itself gives staff an increased administrative workload and with an increased workload there is less time left for being with the patients – so maybe you have created for yourself an environment that lead to more aggression - just some thoughts for you to think about and to sharpen your discussion.

6. PLOS authors have the option to publish the peer review history of their article (what does this mean?). If published, this will include your full peer review and any attached files.

Reviewer #1: No

Reviewer #2: No

---

## [Author Response · Author response to Decision Letter 0]

28 Aug 2020

Please see our cover letter and rebuttal letter for the specific responses to the editor and reviewers comments

---

## [Decision Letter · Decision Letter 1]

22 Sep 2020

Monitoring risk assessment on an acute psychiatric ward: effects on aggression, seclusion and nurse behaviour

PONE-D-19-34513R1

Dear Dr. Florisse,

We’re pleased to inform you that your manuscript has been judged scientifically suitable for publication and will be formally accepted for publication once it meets all outstanding technical requirements.

Kind regards,

Robert Didden

Academic Editor

PLOS ONE

Additional Editor Comments (optional):

Reviewers' comments:

Reviewer's Responses to Questions

**Comments to the Author**

1. If the authors have adequately addressed your comments raised in a previous round of review and you feel that this manuscript is now acceptable for publication, you may indicate that here to bypass the “Comments to the Author” section, enter your conflict of interest statement in the “Confidential to Editor” section, and submit your "Accept" recommendation.

Reviewer #1: All comments have been addressed

Reviewer #2: All comments have been addressed

2. Is the manuscript technically sound, and do the data support the conclusions?

Reviewer #1: Yes

Reviewer #2: Yes

3. Has the statistical analysis been performed appropriately and rigorously? 

Reviewer #1: Yes

Reviewer #2: Yes

4. Have the authors made all data underlying the findings in their manuscript fully available?

Reviewer #1: Yes

Reviewer #2: Yes

5. Is the manuscript presented in an intelligible fashion and written in standard English?

Reviewer #1: Yes

Reviewer #2: Yes

6. Review Comments to the Author

Reviewer #1: Thank you for submitting a revised version of your manuscript to Plos One, and for addressing and responding to the reviewer comments.

Reviewer #2: Thanks again for a well written article. I think you have responded to my comments in an appropriate way.

7. PLOS authors have the option to publish the peer review history of their article (what does this mean?). If published, this will include your full peer review and any attached files.

Reviewer #1: No

Reviewer #2: No

---

## [Editor Report · Acceptance letter]

24 Sep 2020

PONE-D-19-34513R1 

Monitoring risk assessment on an acute psychiatric ward: effects on aggression, seclusion and nurse behaviour 

Dear Dr. Florisse:

I'm pleased to inform you that your manuscript has been deemed suitable for publication in PLOS ONE. Congratulations! Your manuscript is now with our production department. 

Kind regards, 

on behalf of

Professor Robert Didden 

Academic Editor

PLOS ONE